# Pinocembrin Protects from AGE-Induced Cytotoxicity and Inhibits Non-Enzymatic Glycation in Human Insulin

**DOI:** 10.3390/cells8050385

**Published:** 2019-04-26

**Authors:** Margherita Borriello, Clara Iannuzzi, Ivana Sirangelo

**Affiliations:** Department of Precision Medicine, Università degli Studi della Campania “Luigi Vanvitelli”, Via L. De Crecchio 7, 80138 Naples, Italy; margherita.borriello@unicampania.it (M.B.); ivana.sirangelo@unicampania.it (I.S.)

**Keywords:** advanced glycation end products (AGEs), oxidative stress, dicarbonyl stress, glycation, human insulin, AGE–RAGE pathways, pinocembrin

## Abstract

Advanced glycation end products (AGEs) are the end products of the glycation reaction and have a great importance in clinical science for their association with oxidative stress and inflammation, which play a major role in most chronic diseases, such as cardiovascular disease, neurodegenerative diseases, and diabetes. Their pathogenic effects are generally induced by the interaction between AGEs and the receptor for advanced glycation end product (RAGE) on the cell surface, which triggers reactive oxygen species production, nuclear factor kB (NF-kB) activation, and inflammation. Pinocembrin, the most abundant flavonoid in propolis, has been recently proven to interfere with RAGE activation in Aβ–RAGE-induced toxicity. In the present study, we investigated the ability of pinocembrin to interfere with RAGE signaling pathways activated by AGEs. Interestingly, pinocembrin was able to inhibit oxidative stress and NF-kB activation in cells exposed to AGEs. In addition, it was able to block caspase 3/7 and 9 activation, thus suggesting an active role of this molecule in counteracting AGE–RAGE-induced toxicity mediated by NF-kB signaling pathways. The ability of pinocembrin to affect the glycation reaction has been also tested. Our data suggest that pinocembrin might be a promising molecule in protecting from AGE-mediated pathogenesis.

## 1. Introduction

Pinocembrin (5,7-dihydroxyflavone) is a natural flavonoid present at high concentration in propolis, honey, and in several plants and vegetables [1] (Figure 1). Pinocembrin demonstrates several biological and pharmacological properties, including antimicrobial, anti-inflammatory, antioxidant, and anticancer activities [1,2]. In addition, pinocembrin has been shown to possess neuroprotective properties against cerebral ischemic injury through anti-oxidative and anti-inflammatory activities both in vitro and in vivo [2,3,4,5,6,7,8,9]. After oral administration, pinocembrin is well metabolized and absorbed [10], and it is able to pass through the blood–brain barrier through a passive transport process [11]. These characteristics make pinocembrin an attractive potential drug and a suitable molecule for the treatment of central nervous system diseases. Recently, it has been shown that pinocembrin exerts protective effects against Aβ-induced toxicity both in vivo and in vitro mouse and cell models [12,13,14,15,16,17]. Specifically, pinocembrin showed direct inhibition of the receptor for advanced glycation end product (RAGE) and its downstream signaling pathways [12,13]. RAGE is a multiligand transmembrane receptor of the immunoglobulin superfamily expressed in several cell types, and it recognizes different ligands such as advanced glycation end products (AGEs) and Aβ peptide [18,19,20]. RAGE can be abnormally up- or down-regulated in human diseases, and it is highly expressed in an Aβ-rich environment [21,22,23,24]. As a consequence of the Aβ–RAGE interaction, activation of p38 mitogen-activated protein kinases (p38MAPK), stress-activated protein kinase or c-Jun N-terminal kinase (SAPK/JNK), and nuclear factor kB (NF-kB) signaling occur, indicating that RAGE can mediate a common proinflammatory pathway in Alzheimer’s disease [20,21,25]. Irrespective from RAGE involvement in the development and progression of Alzheimer’s disease through its interaction with the Aβ peptide, RAGE is linked to other neurodegenerative and aging-related chronic diseases through its interaction with AGEs [26,27,28]. In fact, owing to their chemical, pro-oxidant, and inflammatory activities, clear evidence points to the involvement of AGEs in degenerative disorders, such as cardiovascular disease, neurodegenerative pathologies, and diabetes complications [29,30,31]. AGEs are a heterogeneous class of compounds resulting from a spontaneous non-enzymatic reaction between reducing sugars or their reactive metabolites, such as methylglyoxal (MG) and proteins, via rearrangement, oxidation, dehydration, and polymerization [32,33]. This process is known as Maillard reaction (MR), and its first step consists of the reaction between reducing sugars and amino groups to reversibly form the Amadori product (AP); from this key compound, the MR can take several pathways up to the formation of AGE species. For instance, in oxidative conditions, AP can be easily converted into intermediate products and promote carbohydrate fragmentation, causing the formation of highly reactive dicarbonyls. These compounds, especially glyoxal and methylglyoxal, react with arginine and lysine residues present in proteins, modifying the lateral amino acid chain and promoting protein cross-linking that strongly alters protein structure, properties, and function [33,34,35]. Small amounts of AGEs are generated in vivo as a normal consequence of metabolism, and in normal physiological conditions, the rapid turnover of intracellular proteins and the action of pathways of AGE detoxification indirectly protect the tissues from high AGE accumulation and its subsequent chemical damage [33]. However, during aging and under pathologic conditions, these defense systems are unable to properly prevent AGE production, leading to their accumulation and triggering cellular toxicity.

Significant efforts have been made to study compounds able to inhibit the glycation reaction or contrast AGE-induced toxicity with the aim of developing new potential therapeutic strategies [36,37,38]. Taking into account the ability of pinocembrin to contrast the RAGE toxicity pathways induced by the Aβ-peptide, the aim of this study is to explore the effect of pinocembrin in AGE-induced toxicity. In addition, the ability of pinocembrin to affect the glycation reaction has also been tested. We have recently reported that insulin glycation by D-ribose strongly affects cell viability through the activation of RAGE pathways. In particular, we have shown that glycated insulin triggers death pathways in different cellular models through the activation of caspases 3 and 7, intracellular reactive oxygen species (ROS) production, and activation of the transcription factor NF-kB [39]. As glycated insulin can be considered to be a good AGE model for the study of the AGE–RAGE activation pathways, in the present study, we have investigated the effect of pinocembrin on the glycation reaction of human insulin, as well as its ability to counteract insulin–AGE-induced toxicity. The results show that pinocembrin strongly affects the insulin glycation reaction kinetics in vitro, and it is able to counteract AGE–RAGE-induced toxicity mediated by NF-kB signaling pathways.

## 2. Materials and Methods

### 2.1. Materials

Human insulin, N-acetyl-L-tyrosine-ethyl ester, methylglyoxal, D-ribose, and 3-(4,5-dimethylthiazol-2-yl)-2,5-diphenyl-tetrazolium bromide (MTT) were used in this study (Sigma-Aldrich Co., St. Louis, MO, USA). Pinocembrin high purity (˃98%) was purchased from Cymit Química S.L. (Barcelona, Spain). The primary antibodies used were as follows: rabbit anti NF-kB p65 (C22B4, Cell Signaling Technology, Inc. Danvers, MA, USA), rabbit anti α-tubulin (ab4074, Abcam, Cambridge, UK), rabbit anti HistoneH1 (ab61177, Abcam), and mouse anti Vimentin (Sigma-Aldrich Co.). The secondary antibodies used were the following: Alexa Fluor 488 and Alexa Fluor 633 (Life Technologies Italia, Monza, Italy). All other chemicals were of analytical grade. Methylglyoxal was further purified by distillation under low pressure, and its concentration was determined spectrophotometrically using ε_284_ = 12.3 M^−1^cm^−1^ [40,41].

### 2.2. Insulin Preparation and Glycation

Human insulin was dissolved in ultra-pure milliQ water to a final concentration of 4 mg/mL at pH 2.0 in order to obtain monomeric insulin [42]; protein concentration was determined by absorbance (ε_275_ = 4560 M^−1^cm^−1^). Finally, insulin was neutralized to pH 7.0 and kept in phosphate buffer 50 mM at pH 7.0. Glycated insulin was prepared mixing human insulin at a final concentration of 0.5 mg/mL and 0.5 mM methylglyoxal or 0.5 M D-ribose in 50 mM NaH_2_PO_4_ buffer at pH 7.0, passed through a 0.22 μm filter, and incubated at 37 °C in sterile conditions for 7 days. Human insulin in buffer without glycating agent was used as the protein control. To test the effect of pinocembrin in insulin glycation kinetics, the same samples were incubated in the absence and in the presence of 0.1, 1, 10, and 20 mM pinocembrin.

### 2.3. Fluorescence Measurements

Fluorescence measurements were performed on a Perkin Elmer Life Sciences LS 55 spectrofluorometer (Perkin Elmer, Waltham, MA, USA). Tyrosine fluorescence emission (λ_ex_ 275 nm/λ_em_ 305 nm) was evaluated on both insulin and free tyrosine after the addition of pinocembrin at different insulin/pinocembrin molar ratios (1:0, 1:0.5, 1:0.75, and 1:1). Tyrosil fluorescence quenching was monitored by estimation of the F_0_/F ratio considering the fluorescence intensity at 305 nm of the sample before (F_0_) and after (F) the addition of pinocembrin. Working concentrations were 10 µM for insulin and 40 µM for free tyrosine. To assess the intrinsic fluorescence of AGEs (λ_ex_ 320 nm/λ_em_ 410 nm), glycated insulin at a final concentration of 8 μM was monitored at different incubation times with the glycating agent in the absence and in the presence of pinocembrin. The fluorescence signal of pinocembrin at both excitation wavelengths (275 and 350 nm) was negligible (Appendix A). The fluorescence intensity was corrected by subtracting the emission intensity of d-ribose/methylglyoxal solutions at different incubation times [43].

### 2.4. Cell Cultures and Treatments

CPAE endothelial cells (ECs) (ATCC#CCL-209) were cultured in minimum essential medium (MEM) supplemented with 10% fetal bovine calf serum (USA Origin), 2.0 mM glutamine, 100 units/mL penicillin, and 100 mg/mL streptomycin in a 5.0% CO_2_ humidified environment at 37 °C. Cells were exposed to ribose-glycated insulin for 7 days (final concentration 40 µM) in the presence and in the absence of 40 µM pinocembrin. The ribose-glycated insulin was extensively dialyzed in sterile conditions against phosphate buffer 50 mM at pH 7.0 before incubation with cell culture to remove the free glycating agent. For all experiments, untreated cells and cells incubated in the presence of only pinocembrin at the tested concentrations served as the control.

### 2.5. MTT Assay

Cell viability was assessed as the inhibition of the ability of cells to reduce the metabolic dye 3-[4–dimethylthiazol-2-yl]-2,5-diphenyltetrazolium bromide (MTT) to a blue formazan product [44]. After indicated times of incubation with protein samples, cells were rinsed with phosphate buffer solution (PBS). A stock solution of MTT (5 mg/mL in PBS) was diluted 10 times in cell medium and incubated with cells for 3 h at 37 °C. After removing the medium, cells were treated with isopropyl alcohol and 0.1 M HCl for 20 min. Levels of reduced MTT were assayed by measuring the difference in absorbance between 570 and 690 nm. Data are expressed as average percentage reduction of MTT with respect to the control. In the cell proliferation assay, the MTT absorbance was evaluated at 570 nm.

### 2.6. Cell Cycle Analysis

After 48 h of incubation with protein samples (40 μM), 2.5 × 10^5^ cells were collected and resuspended in 500 μL of hypotonic buffer (0.1% Triton X-100, 0.1% sodium citrate, 50 μg/mL iodide propidium, RNAse A). Cells were incubated in the dark for 30 min, and samples were acquired on a FACSCalibur flow cytometer using Cell Quest software (Becton Dickinson, San Jose, CA, USA) and ModFitLT version 3 software (Verity Software House, Topsham, ME) as previously reported [45].

### 2.7. Detection of Intracellular ROS

Intracellular ROS were detected by means of an oxidation-sensitive fluorescent probe, 2′,7′-dichlorofluorescin diacetate (DCFH-DA). Cells were grown in 12-well plates, pre-incubated with DCFH-DA for 30 min, and then incubated with protein samples for 24 h. Control experiments were performed using untreated cells and cells exposed to a 0.001 M H_2_O_2_. After incubation, cells were washed twice with PBS buffer and then lysed with Tris-HCl 0.5 M, pH 7.6, 1% SDS. The non-fluorescent DCFH-DA was converted, by oxidation, to the fluorescent molecule 2′,7′-dichlorofluorescein (DCF). DCF fluorescence intensity was quantified on a Perkin Elmer Life Sciences LS 55 spectrofluorometer (Perkin Elmer) using an excitation wavelength of 488 nm and an emission wavelength of 530 nm [46].

### 2.8. Confocal Laser-Scanning Microscopy

Confocal microscope analysis was performed as previously described [38]. Briefly, cells were fixed with 3% paraformaldehyde and permeabilized with 0.1% Triton X-100 before incubation with specific antibodies against NF-kB (1:100, rabbit) and Vimentin (1:1.000, mouse). Secondary antibodies were Alexa Fluor 488 (1:1000) or Alexa Fluor 633 (1:1000). Microscopy analyses were performed using a Zeiss LSM 700 confocal microscope (Carl Zeiss MicroImaging, Jena, Germany) equipped with a plan-apochromat X 63 (NA 1.4) oil immersion objective (Zeiss, Thornwood, NY, USA).

### 2.9. Cellular Nuclear Extraction

Control and treated cells (1 × 10^6^ cells) were pelleted by centrifugation, resuspended in lysis buffer (10 mM HEPES pH 7.5, 10 mM KCl, 0.1 mM EDTA, 1 mM dithiothreitol, 0.5% Nonidet-40, and 0.5 mM PMSF, along with the protease inhibitor cocktail) and allowed to swell on ice for 15–20 min. Tubes were vortexed to disrupt cell membranes and then centrifuged at 12,000× *g* at 4 °C for 10 min. The supernatant was taken as cytoplasmic extract. The pelleted nuclei were washed thrice with the cell lysis buffer, resuspended in the nuclear extraction buffer (20 mM HEPES (pH 7.5), 400 mM NaCl, 1 mM EDTA, 1 mM DTT, 1 mM PMSF with protease inhibitor cocktail), and incubated on ice for 30 min. Nuclear extract was collected by centrifugation at 12,000× *g* for 15 min at 4 °C. Protein concentration of the nuclear and cytoplasmic extract was estimated using Bradford’s reagent (BioRad, Hercules, CA, USA). Cytoplasmic contamination of the nuclear fraction was tested by checking tubulin through western blot analysis.

### 2.10. Immunoblotting

Proteins were separated by 10% SDS-PAGE under reducing conditions and blotted onto a polyvinylidene difluoride membrane in transfer buffer (25 mM Tris, 192 mM glycine, 20% methanol, 0.1% SDS). The blots were then probed with primary antibodies, followed by the corresponding horseradish peroxidase (HRP)-conjugated secondary antibodies. Immunoreactivity was detected by the ECL reaction (RPN2109, GE Healthcare, Chicago, IL, USA) and quantified using the ChemiDoc MP Imager Software (version 2.0, Biorad).

### 2.11. Caspase Assay

Caspase activity was detected within living cells using B-BRIDGE Kits (As One International, Santa Clara, CA USA), supplied with cell-permeable fluorescent substrates, following the manufacturer’s suggestions. The fluorescent substrates used were FAM-DEVD-FMK for caspase-3/7 and SR-LEHD-FMK for caspase 9. After 24 h of incubation with protein samples, cells were collected, washed twice in cold PBS, and incubated for 1 h on ice with the substrate. Cells were analyzed using Cell Quest software applied to a FACSCalibur (Becton Dickinson).

### 2.12. Statistical Analysis

Statistical analyses were performed using Stata software (Version 13.0; StataCorp LP., College Station, TX, USA). Tukey’s post hoc test was used if the treatment was significant on analysis of variance (ANOVA). All data are represented as the mean ± SE. Statistical significance was set at *p* < 0.05.

## 3. Results

### 3.1. Pinocembrin Effect on AGE-Induced Toxicity

Insulin is susceptible to glycation by glucose, d-ribose, and other highly reactive carbonyls, such as methylglyoxal, especially in diabetic conditions, and the AGE products are considered to be the main cause of diabetes-related vascular complications [47,48,49]. We have recently reported that insulin glycation by D-ribose produces AGEs adducts that strongly affect the cell viability in different cellular models [39]. In order to test the ability of pinocembrin to reduce AGE toxicity, we evaluated the cell viability in cells exposed to fully glycated species in the presence and in the absence of pinocembrin. In particular, we performed both MTT viability assay and cell cycle evaluation in CPAE ECs co-incubated with pinocembrin and ribosylated insulin in a 1:1 molar ratio (Figure 2). The viability assay was performed at 0, 24, and 48 h of incubation with insulin–AGE (Figure 2A). As expected, at 24 h of treatment, glycated insulin induced a strong reduction of the cell viability (68%), whereas in the presence of pinocembrin only a 20% reduction was observed. Similarly, flow cytometry analysis showed that glycated insulin was able to significantly alter the cell cycle at 48 h of treatment, whereas in the presence of pinocembrin, no appreciable changes in cell cycle were observed (Figure 2B). Specifically, the ribose-glycated insulin sensitized cells to death with an increase of pre-G1 phase from 2% (CTR cells) to 25%, whereas in the presence of pinocembrin no alteration of the cell cycle distribution was observed. These data suggest that pinocembrin, when co-incubated with glycated insulin, is able to protect cells from insulin–AGE toxicity.

### 3.2. Pinocembrin Effect on AGE-Induced Oxidative Stress, NF-kB, and Caspase Activation

As AGE toxicity is generally associated with oxidative stress, we first tested the ability of pinocembrin to reduce the AGE-induced ROS production [39,45]. To this aim, we measured the intracellular ROS levels in ECs co-incubated with ribosylated insulin and 40 µM pinocembrin (1:1 molar ratio) for 24 h, by DCFH-DA fluorescence assay (Figure 3).

Interestingly, while ribosylated insulin promotes ROS production as indicated by the increase in the DCF fluorescence, the sample co-incubated with pinocembrin showed a strong reduction of ROS levels. These data suggest that pinocembrin exerts a protective effect in AGE-induced cytotoxicity, likely affecting death pathways mediated by intracellular ROS production.

AGEs are generally responsible, via AGE–RAGE interaction, for an increase in oxidative stress and inflammation through the formation of ROS and the activation of the NF-kB [50,51]. Activated NF-kB transfers from cytoplasm to the nucleus and regulates the transcription of several target genes. Glycated insulin is known to promote oxidative stress and NF-kB and caspase 3/7 activation through AGE–RAGE pathway signaling [38]. In order to better analyze the protective effect observed for pinocembrin in AGE-related toxicity and ROS production, we evaluated the NF-kB activation in ECs exposed to glycated insulin in the absence and in the presence of pinocembrin (Figure 4). In particular, a confocal immunofluorescent assay was performed on ECs incubated for 24 h with glycated insulin in the presence and in the absence of pinocembrin (Figure 4A–D). As expected, exposure of cells to glycated insulin resulted in NF-kB activation, as depicted by its immunofluorescence signal (Figure 4C), whereas no activation of NF-kB was observed in cells co-incubated with pinocembrin (Figure 4D). These results are consistent with western blot analysis, which showed no translocation of NF-kB/p65 to the nucleus in the presence of pinocembrin (Figure 4E). At the same time, no activation of caspase 3/7 and caspase 9 was observed in ECs co-incubated with glycated insulin and pinocembrin (Figure 5). Taken together these data suggest that pinocembrin is able to counteract AGE-related toxicity by directly affecting ROS production and NF-kB and caspase activation.

### 3.3. Pinocembrin Effect in Insulin Glycation Reaction

Insulin is susceptible to glycation by glucose, d-ribose, and other highly reactive carbonyls, such as methylglyoxal, especially in diabetic conditions [39,47,49]. Indeed, both d-ribose and methylglyoxal have been shown to efficiently react with human insulin, producing fully glycated species in few days [39]. To evaluate the effect of pinocembrin in insulin glycation kinetics, we performed the glycation reaction in the presence of pinocembrin, and we monitored AGE formation by fluorescence spectroscopy as AGEs are characterized by a typical fluorescence emission at 410 nm upon excitation at 320 nm [52]. To this aim, insulin samples were incubated at 37 °C with 0.5 M D-Ribose or 0.5 mM MG in the presence of 10 mM pinocembrin, and AGE fluorescence was monitored in time (Figure 6).

For both D-Ribose and MG, in the absence of pinocembrin, the emission intensity at 410 nm increased markedly with incubation time, and the glycation reaction was completed in about 7 days. Differently, in the presence of pinocembrin, a drastic reduction of AGE formation was detected at all incubation times, as indicated by the decrease of the fluorescence intensity.

In addition, to evaluate a concentration-dependent effect of pinocembrin in the glycation reaction, insulin samples were incubated at 37 °C with 0.5 M d-Ribose or 0.5 mM MG at different concentrations of pinocembrin (0, 0.1, 1, 10, and 20 mM), and AGE fluorescence was monitored after 7 days (Figure 7). The inhibition of AGE formation was stronger at increasing concentrations of pinocembrin in the presence of both d-Ribose and MG, and 1mM pinocembrin was enough to strongly restrain the process. These data indicate that pinocembrin strongly inhibits insulin glycation by d-Ribose and MG in a concentration-dependent manner.

### 3.4. Characterization of Insulin–Pinocembrin Interaction

The pinocembrin–insulin interaction was investigated by fluorescence spectroscopy in native conditions. Insulin contains four tyrosyl residues as fluorescence emitters, and its spectrum is characterized by the typical tyrosyl emission centered at 305 nm. The emission fluorescence spectra of insulin in the absence and in the presence of pinocembrin at different molar ratios are shown in Figure 8A. The fluorescence intensity regularly decreased upon the addition of increasing concentrations of pinocembrin, thus indicating that the insulin–pinocembrin interaction induces quenching of tyrosine emission. The fluorescence quenching of a protein caused by small molecules may be collisional or due to the formation of a complex that has zero or small quantum yield. In order to exclude a collisional quenching by pinocembrin, we performed the same experiment on the monomeric tyrosyl residue (*N*-acetyl-l-tyrosine ethyl ester) (Figure 8B). The F_0_/F values recorded for free tyrosine were significantly lower than those recorded for insulin at each molar ratio, indicating the formation of an insulin–pinocembrin complex, as no collisional quenching is involved. The ability of pinocembrin to induce conformational changes in human insulin was explored by far-UV CD spectroscopy, and no difference in the secondary structure content was observed in the presence of pinocembrin (data not shown). Moreover, as no change in the emission maximum was observed upon the addition of pinocembrin (Figure 8A), we can hypothesize that no strong conformational changes occur in human insulin in the presence of pinocembrin.

The binding of pinocembrin to human insulin could be responsible for the inhibition of the glycation reaction as pinocembrin could interact with insulin regions in close proximity to the glycation sites (Lys and Arg residues). To validate this hypothesis, we performed the glycation reaction with d-Ribose or MG both in the presence and in the absence of pinocembrin and monitored the tyrosine emission fluorescence at the end of the process (7 days) (Figure 9). Interestingly, the tyrosyl fluorescence was strongly reduced in the glycated protein, suggesting that pinocembrin might induce small conformational changes in the tyrosyl environment that could reduce accessibility to the close glycation sites.

## 4. Discussion

Pinocembrin (5,7-dihydroxyflavanone), a natural flavonoid found at high concentration in propolis, is known to possess several pharmacological activities, such as anti-inflammatory, anti-oxidant, and anti-cancer properties [1,2]. Moreover, a therapeutic action in cognitive function and neuronal protection for pinocembrin has been recently suggested [3,4,5,6,7,8,9,53]. Specifically, it has been reported that pinocembrin exerts an active role in neuronal protection against Aβ-induced toxicity through specific inhibition of the Aβ–RAGE-mediated signaling pathways. In particular, pinocembrin significantly inhibits the upregulation of RAGE transcripts and protein expression both in vivo and in vitro [12,16]. In view of this observation, in our study we analyzed the ability of pinocembrin to interfere with the AGE–RAGE signaling pathways. Indeed, although several ligands are able to bind RAGE, its activation is strongly mediated by AGE species in vivo [51]. These species are the end products of the nonenzymatic glycation reaction, and their importance in clinical science is strictly related to oxidative stress and inflammation—common processes in many diseases, including cardiovascular disease, ischemic injury, diabetes, cancer, and neurodegenerative diseases [29,30].

In order to investigate if pinocembrin was able to protect cells from AGE toxicity through the inhibition of RAGE signaling pathways, we tested its effect using glycated insulin as an AGE model. In our previous study, we showed that insulin glycated by D-ribose strongly affects cell viability, triggering death pathways mediated by the activation of RAGE, NF-kB, caspases 3/7 and 9, and intracellular ROS production [38]. Interestingly, our results suggest that pinocembrin also exerts a protective effect in AGE-induced cytotoxicity. In particular, both cell viability and cell cycle analysis indicated that, in endothelial cells exposed to glycated insulin, pinocembrin was able to restore cell cycle distribution and viability. Moreover, pinocembrin showed a marked antioxidant activity also in AGE-induced oxidative stress.

Pinocembrin has been recently shown to suppress inflammatory response by blocking the activation of the transcription factor NF-kB [13,54,55]. NF-kB regulates many cellular pathways and processes, including inflammation, immune response, apoptosis, and proliferation, and it is also involved in AGE-induced pathogenesis as persistent activation of NF-kB has been observed in the AGE–RAGE signaling pathways [50,51,55,56]. For this reason, the effect of pinocembrin in the activation of NF-kB associated with the AGE–RAGE pathways has been evaluated in cells exposed to glycated insulin. Interestingly, pinocembrin was able to inhibit the NF-kB translocation into the nucleus induced by glycated insulin. In addition, pinocembrin was able to block caspase 3/7 and 9 activation, thus suggesting an active role of this molecule in counteracting the AGE–RAGE-induced toxicity mediated by NF-kB signaling pathways. A similar effect has been observed for pinocembrin in the Aβ–RAGE-mediated toxicity [12,13,14]. In particular, in both Aβ-treated mice and cellular models, pinocembrin was shown to restore cell viability by preventing RAGE upregulation through a direct inhibition of MAPK-kinase2, NF-kB, caspase 3/7, and caspase 9 [12,13]. In this respect, we can hypothesize a similar protection mechanism for pinocembrin from both Aβ–RAGE and AGE–RAGE-induced toxicity.

Besides triggering intracellular signaling via RAGE, AGEs can also induce pathology in a receptor-independent manner. Indeed, AGE formation involves protein cross-linking that strongly alters protein structure, properties, and function [31,32]. As several flavonoids are known to restrain the glycation reaction, the effect of pinocembrin was tested in the glycation of human insulin. Human insulin is known to be glycated at Arg22, Lys29, and N-terminus [47,57]. Very interestingly, pinocembrin was able to strongly inhibit the reaction in the presence of two different glycating agents, d-ribose and methylglyoxal, in a concentration-dependent manner. Such inhibition could be due to the interaction between pinocembrin and regions in close proximity to the glycation sites in human insulin as suggested by the intrinsic fluorescence results. Tyrosyl residues in human insulin are in close proximity to the glycation sites (Lys29 and Arg22), and their fluorescence was strongly reduced in the presence of pinocembrin. We can hypothesize that pinocembrin might induce small conformational changes in the tyrosyl environment that could reduce accessibility to the close glycation sites. However, further studies will be needed to identify the molecular mechanism of pinocembrin action. Similarly, it has been reported that several flavonoids are able to bind albumin in close proximity to its glycation sites, thus enhancing their protective efficiency in glycoxidation [58]. Flavonoids are the group of polyphenols with the highest potential for the inhibition of glycoxidation, and this effect is supposed be mainly due to their antioxidant properties [59]. Glycation can be catalyzed by metals, and it is associated with the generation of ROS and oxidation; the combination of both processes is often referred to as glycoxidation. For this reason, capturing free radicals to reduce both oxidative stress and production of reactive carbonyls in the glycation process can strongly affect AGE formation [38]. Indeed, oxidative conditions are known to be a key factor in promoting protein glycation and AGE formation in vivo. In this respect, we can hypothesize that the anti-glycating action of pinocembrin could be ascribed both to lower accessibility of glycation sites upon its interaction with insulin and to its antioxidant activity able to reduce the production of reactive carbonyls—key species in the glycation process.

Due to the relevant clinical/pathological impact, great efforts are currently being undertaken to identify molecular targets able to prevent or limit AGE-dependent cell damage, including the use of synthetic or natural AGE inhibitors and compounds able to reduce inflammation and oxidative stress. For this reason, therapeutic intervention strategies aimed at inhibiting AGE-mediated complications should involve targeting either AGE formation or AGE-induced toxicity. Although several synthetic compounds can efficiently inhibit AGE formation or break protein cross-links, they can also have severe side effects. For this reason, recently, much attention has been paid to natural compounds, such as polyphenols, well known for their antioxidants and anti-inflammatory properties [60]. The ability of polyphenols to suppress methylglyoxal and AGE formation has been reported and also confirmed in in vivo studies [37]. In addition, several studies have suggested the combined use of two or more polyphenols in order to enhance the beneficial properties of the single components. In this respect, considering its combined anti-glycating, antioxidant, and anti-inflammatory properties, pinocembrin might be a promising molecule in protecting against AGE-induced cytotoxicity.

## Figures and Tables

**Figure 1 cells-08-00385-f001:**
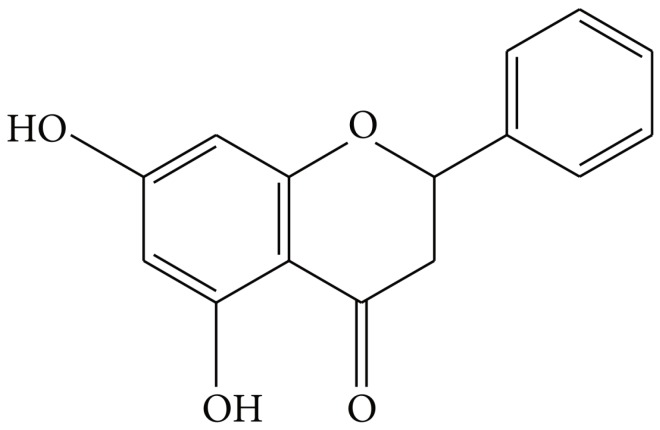
Chemical structure of pinocembrin.

**Figure 2 cells-08-00385-f002:**
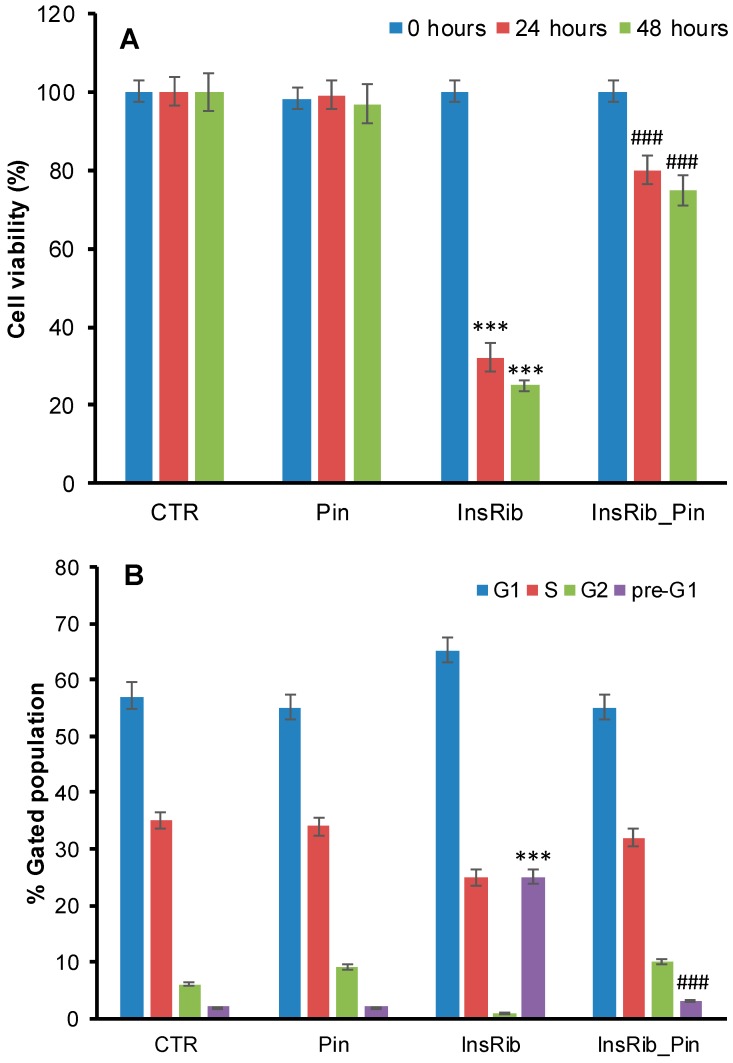
Effect of pinocembrin in AGE-induced toxicity. The cell viability was evaluated by 3-(4,5-dimethylthiazol-2-yl)-2,5-diphenyl-tetrazolium bromide (MTT) assay (**A**) and cell cycle analysis (**B**). (**A**) Cell viability evaluated by MTT assay in endothelial cells (ECs) exposed for 0, 24, and 48 h to glycated insulin in the presence (InsRib_Pin) and in the absence (InsRib) of pinocembrin. (**B**) Cell cycle distribution of ECs exposed to glycated insulin for 48 h in the presence (InsRib_Pin) and in the absence (InsRib) of pinocembrin. Bar charts represent the percentage of cell populations. CTR: untreated cells, Pin: cells treated with 40 µM pinocembrin. Working concentrations were 40 µM both for glycated insulin and pinocembrin. Other experimental details are described in Section 2. The data are presented as mean ± SE of three replicates in five independent experiments. *** *p* ˂ 0.001 versus CTR, ### *p* ˂ 0.001 versus InsRib.

**Figure 3 cells-08-00385-f003:**
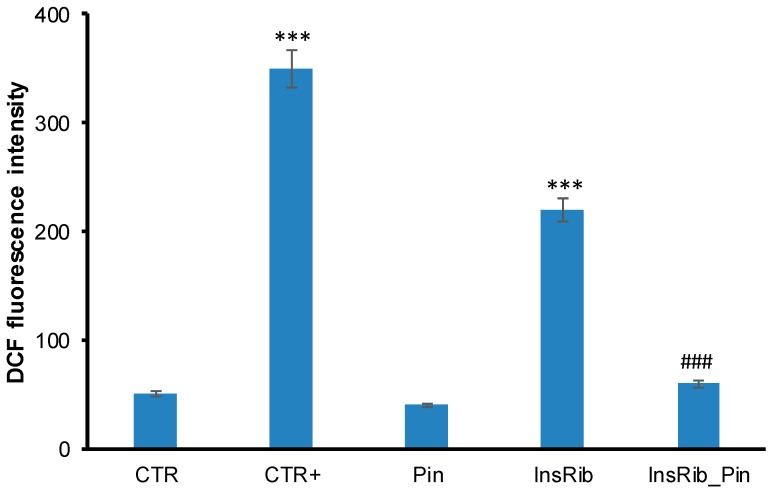
Effect of pinocembrin in advanced glycation end product (AGE)-induced oxidative stress. ECs were exposed in the absence and in the presence of pinocembrin to glycated insulin, and the reactive oxygen species (ROS) production was evaluated after 24 h by 2′,7′-dichlorofluorescin diacetate (DCFH-DA) assay. CTR: untreated cells, CTR+: cells treated with 1.0 mM H_2_O_2_, Pin: cells treated with 40 µM pinocembrin; InsRib: cells treated with glycated insulin, InsRib_Pin: cells co-incubated with glycated insulin and pinocembrin. Working concentrations: 40 µM for insulin and 40 µM for pinocembrin. Other experimental details are described in Section 2. The data are presented as mean ± SE of three replicates in five independent experiments. *** *p* ˂ 0.001 versus CTR, ### *p* ˂ 0.001 versus InsRib.

**Figure 4 cells-08-00385-f004:**
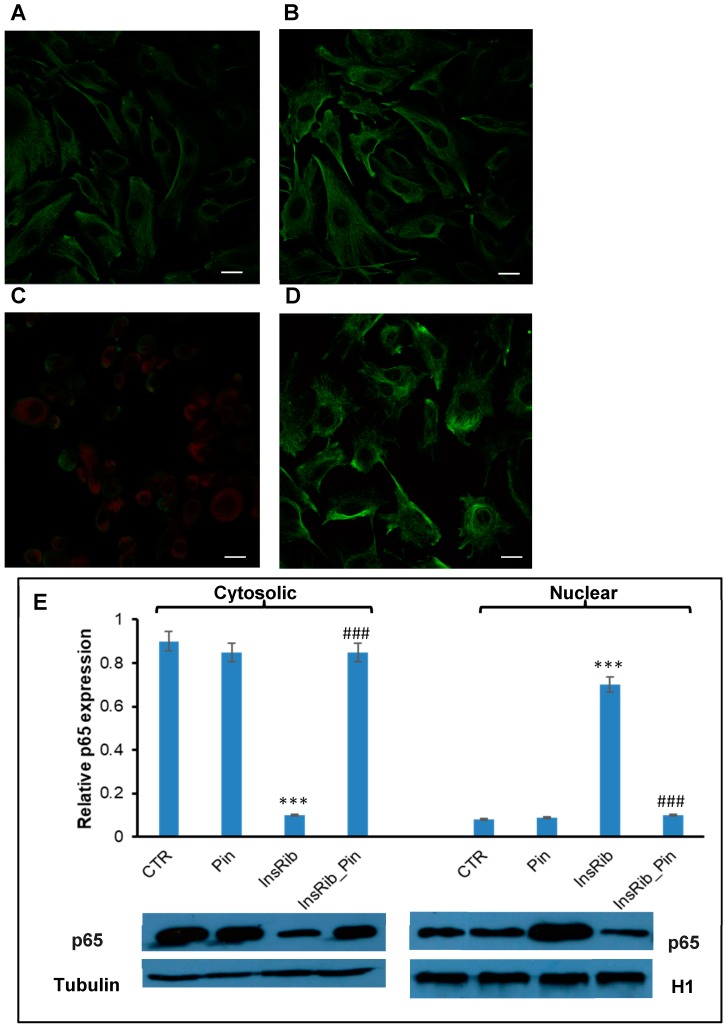
Effect of pinocembrin in nuclear factor kB (NF-kB) activation. Representative confocal images of EC untreated (**A**) and treated for 24 h with pinocembrin (**B**), glycated insulin (**C**), and glycated insulin in the presence of pinocembrin (**D**). Cells were incubated with specific antibodies against NF-kB/p65 (red) and Vimentin (green). Scale bar represents 20 μm. (**E**) Western blot analysis of NF-kB/p65 expression for cytosolic and nuclear fractions. Working concentrations were 40 µM both for insulin and pinocembrin. CTR: untreated cells, Pin: cells treated with 40 µM pinocembrin, InsRib: cells treated with glycated insulin, InsRib_Pin: cells co-incubated with glycated insulin and pinocembrin. Other experimental conditions are described in Section 2. Data are means ± SE (*n* = 3). *** *p* ˂ 0.001 versus CTR, ### *p* ˂ 0.001 versus InsRib.

**Figure 5 cells-08-00385-f005:**
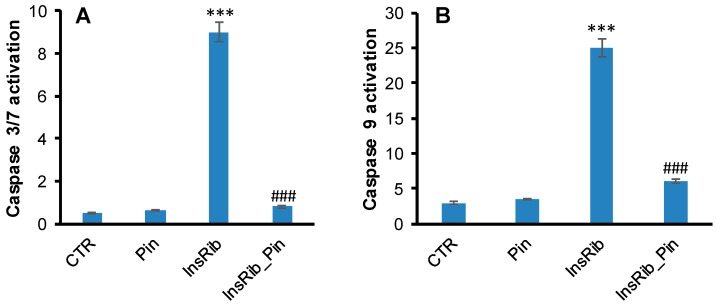
Effect of pinocembrin in caspase 3/7 (**A**) and caspase 9 (**B**) activation. Working concentrations were 40 µM for both insulin and pinocembrin. CTR: untreated cells, Pin: cells treated with 40 µM pinocembrin, InsRib: cells treated with glycated insulin, InsRib_Pin: cells co-incubated with glycated insulin and pinocembrin. Other experimental conditions are described in Section 2. The data are presented as mean ± SE of three replicates in five independent experiments. *** *p* ˂0.001 versus CTR, ### *p* ˂ 0.001 versus InsRib.

**Figure 6 cells-08-00385-f006:**
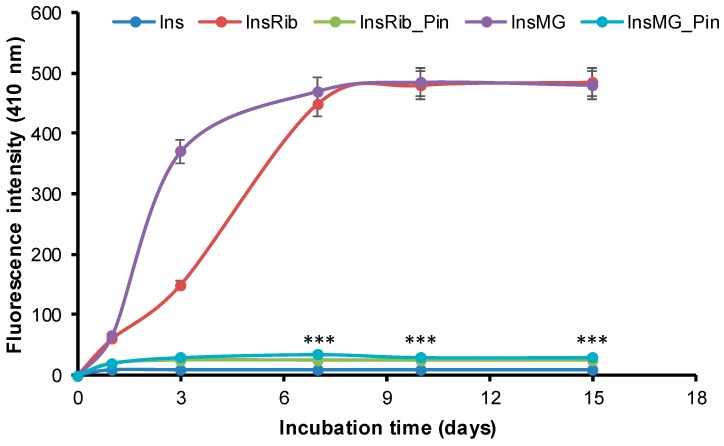
Effect of pinocembrin in insulin glycation kinetics. Insulin samples were incubated at 37 °C with 0.5 M d-ribose (InsRib) and 0.5 mM methylglyoxal (InsMG) in the absence and in the presence of 10 mM pinocembrin and AGE fluorescence (λex 320 nm/λem 410 nm) was monitored at different time points. Ins: Insulin incubated in the absence of glycating agent. Other experimental details are described in Section 2. Data are means ± SE (*n* = 5). *** *p* ˂ 0.001 for InsRib_Pin versus InsRib and InsMG_Pin versus InsMG.

**Figure 7 cells-08-00385-f007:**
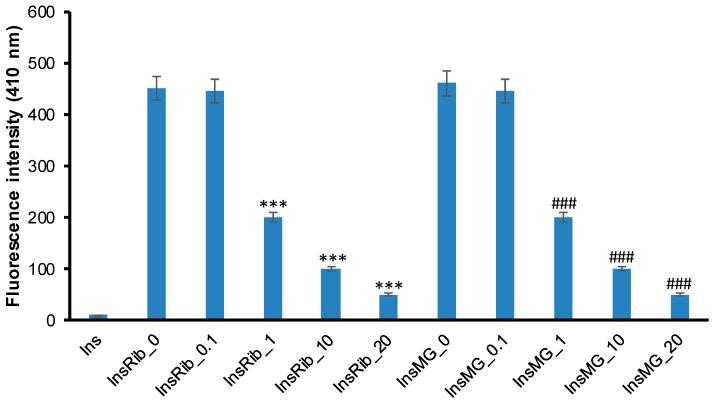
Effect of pinocembrin concentration in insulin glycation kinetics. Insulin samples were incubated at 37 °C with 0.5 M d-ribose (InsRib) or 0.5 mM methylglyoxal (InsMG) at different concentrations of pinocembrin (0, 0.1, 1, 10, and 20 mM) and AGE fluorescence (λex 320 nm/λem 410 nm) was monitored after 7 days of incubation. Ins: Insulin incubated in the absence of glycating agent. Other experimental details are described in Section 2. Data are means ± SE (*n* = 5). *** *p* ˂ 0.001 versus InsRib, ### *p* ˂ 0.001 versus InsMG.

**Figure 8 cells-08-00385-f008:**
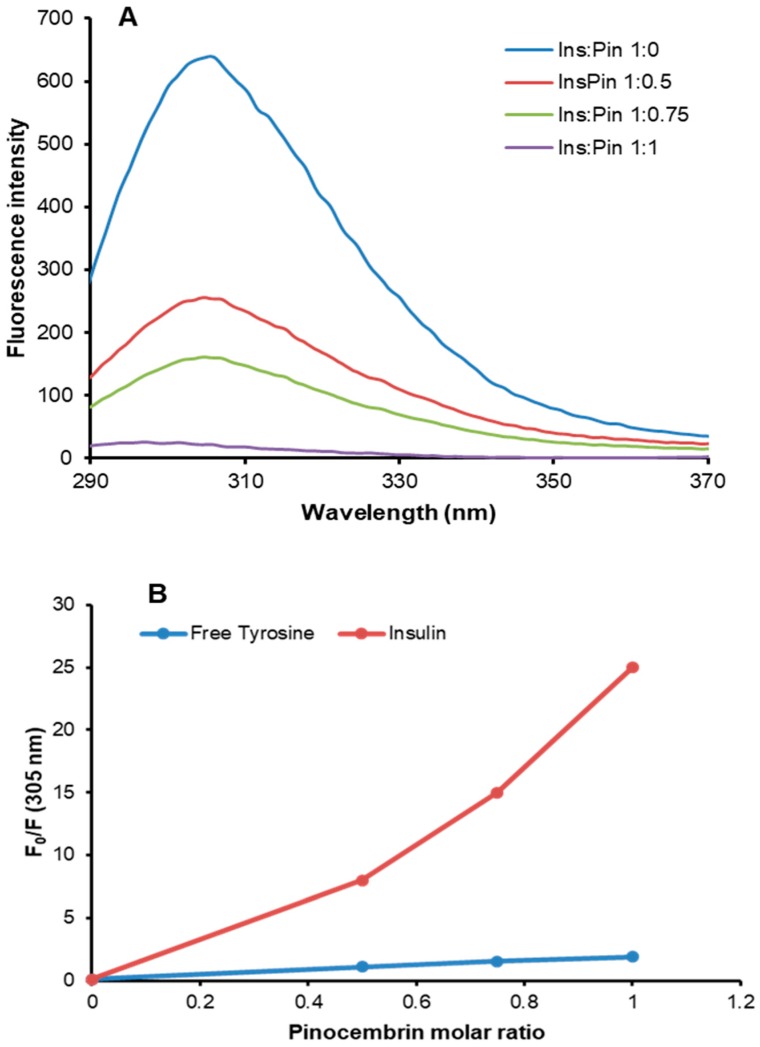
Insulin–pinocembrin interaction monitored by fluorescence spectroscopy. Tyrosine fluorescence emission was evaluated on both insulin (**A**) and free tyrosine after the addition of pinocembrin at different insulin/pinocembrin and tyrosine/pinocembrin molar ratios (1:0, 1:0.5, 1:0.75, and 1:1). (**B**) The dependence of F0/F on the pinocembrin/insulin and pinocembrin/tyrosine molar ratios is shown. Working concentrations were 10 µM for insulin and 40 µM for free tyrosine. Other experimental details are described in Section 2.

**Figure 9 cells-08-00385-f009:**
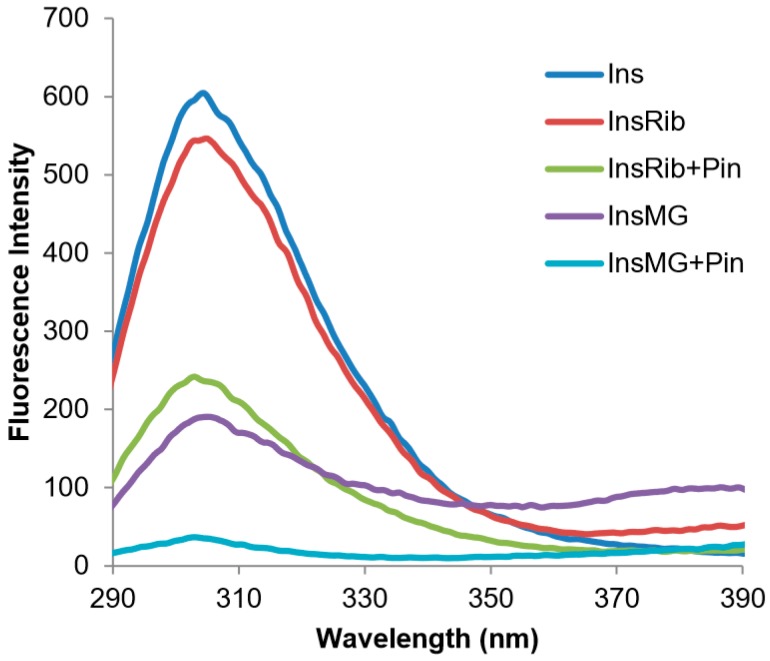
Glycated insulin–pinocembrin interaction monitored by fluorescence spectroscopy. Tyrosine fluorescence emission was evaluated on in native insulin (Ins) and in insulin glycated by d-ribose (InsRib) and methylglyoxal (InsMG) in the presence and in the absence of pinocembrin. Working concentrations were 10 µM for insulin and 10 µM for pinocembrin. Other experimental details are described in Section 2.

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
