# Peer review of "Pinocembrin Protects from AGE-Induced Cytotoxicity and Inhibits Non-Enzymatic Glycation in Human Insulin"

_cells, 2019, doi:10.3390/cells8050385_

Round 1
Reviewer 1 Report
The authors correctly respond to my queries.
Reviewer 2 Report
The authors have sufficiently clarified all issues that was raised. The authors have also given enough elaboration on the result section which were of initial concerns. The graphs have been adequately adjusted. No further quiries are needed for this manuscript and I feel this work is ready for publication
Reviewer 3 Report
The authors have addressed all the points raised in my previous report and clarified all the points according to my request.
This manuscript is a resubmission of an earlier submission. The following is a list of the peer review reports and author responses from that submission.
Round 1
Reviewer 1 Report
This manuscript studies the effect of pinocembrin as inhibitor of the cytotoxicity induced by glycated insulin. The data that the authors provide cloud be of interest, since it suggests that pinocembrin could be used as inhibitor of the protein glycation but also as inhibitor of the cell toxicity induced by the glycation end-products. However, their experimental results (and therefore their findings) need to be further validated by performing additional controls. Without them, their data interpretation is only speculative and this manuscript cannot be accepted for publication.
Major issues
1.- The data provided in Fig 1 (both in panels A and B), in Fig. 2 and in Fig. 3D,E must be complete by adding the results arising from the use of ribose alone and those arising from the use of ribose + pinocembrin.
2.-In addition, the confocal images must be complemented by an additional image showing the cells treated only with ribose.
3.-The authors suggest that pinocembrin could inhibit glycation by blocking the Lys and Arg through a binding process. To validate their hypothesis, the authors must use fluorescence spectroscopy to study whether pinocembrin is also able to inhibit (or not) the AGEs formation from a solution containing Lys and Arg (at the same ratios than present in the sequence of insulin) and ribose/methyglyoxal.
4.-In lines 299-302, the authors state that fluorescence of glycated insulin was reduced upon adding pinocembrin, which could be due to a direct competition between the glycation reagents and pinocembrin. This data interpretation is nonsense, since glycation occurs through a non-reversible covalent modification and once it is done, the AGEs cannot be displaced by any other ligand. The authors should re-write all their interpretation arising from these results.
5.-The authors used a methyglyoxal solution purchased from Sigma, which then purified by distillation under low pressure. Afterwards, they determined its concentration by UV/Vis absorbance at 284nm. To do that, they took a ε of 12.3M-25px-1, which it seems it was taken from ref 37 (i.e. Oya et al. J. Biol. Chem. 1999 274, 18492-502). However, in this reference, the concentration of methylglyoxal was determined using a derivative HPLC method, which involved the use of 2-methylquinoxaline (Anal. Biochem, 1996, 234, 221-224). The authors should correct this in the manuscript, but also explain how they find out that the ε of methylglyoxal at 284nm is 12.3M-25px-1.
6.-In addition, are they sure that the purified methylglyoxal is pure? They provide experimental evidences proving its purity. For example, a 1H-NMR spectrum.
7.-The authors used fluorescence spectroscopy to study the quenching effect of pinocembrin on insulin, which could be attributed to their binding. However, to further validate their results, the authors should provide a control of the fluorescent spectra (λexc 275nm) of pinocembrin under the concentrations that they used in the quenching experiments. In addition, they should do the same, but using a λexc 320nm.
Minor issues
1.-The authors should include an image with the structural formula of pinocembrin. This would help the reader to understand the chemical features of the inhibitor that they are using.
2.-Insulin is a heterodimeric protein whose monomers are covalently linked through the formation of different disulfide bridges. Therefore, it is not clear that insulin can be monomeric at pH 2, unless the authors include a reference supporting this and/or add experimental evidences. In addition, I do not understand why the authors want to obtain monomeric insulin, since this would create a heterogeneous solution containing two different proteins. Moreover, I do not understand the strategy that the authors used to prepare the insulin solutions as afterwards, they neutralized the insulin solutions to pH 7.0. Is insulin dimeric or monomeric at pH 7? How can the authors be sure that the disulfide bridges are properly formed?
3.-The authors should explain in the figure legend of Fig. 1, what does it means G1, S, G2 and pre-G1 (in panel B).
Author Response
Enclosed please find the file with point by point response.

Reviewer 2 Report
In this paper, Borriello et al. report the effects of pinocembrin, a natural flavonoid of vegetal origin, on the insulin glycation by physiological sugars, i.e. D-ribose and methylglyoxal.
The results here reported are interesting since, despite pinocembrin is considered a potential new drug due to its protective effects against some relevant pathologies, no details about its mechanism of action exist.
The paper is well conceived and well written, the experimental work is sound, the results are clearly exposed and supportive of conclusions. My overall evaluation is positive, I only suggest to expand the introduction and the discussion to address some other points which, in my opinion, could improve the impact of the paper. Some main points are raised below.
1. It is not clear to me if, when measuring the influence of “glycated insulin” on cellular viability, ROS production etc., the authors are using insulin previously glycated or the glycation occurs during the experiment itself. As far as I understood, it seems that in all the experiments performed the protein (i.e. native insulin) is incubated with the glycating agent in the presence and in the absence of pinocembrin. Since data showed in figures 4 and 5 indicate that the glycation is strongly inhibited by pinocembrin, as also assessed by the authors, I wonder if it is possible to calculate or at least estimate the amount of glycated protein formed, or at least comment this aspect. Also, based on the previous considerations (i.e. only a small fraction of insulin is glycated in the presence of pinocembrin), the authors should indicate the insulin incubated with glycating agents in the presence of pinocembrin with a different name, since in my opinion “glycated insulin” indicated the product of the reaction between insulin and glycating agents, in the absence of pinocembrin, i.e. extensively glycated, and probably modified at different sites.
2. The authors suggest that, upon binding of pinocembrin to tyrosine side-chains, glycation of lysine and arginine residues is inhibited, but they do not explain how this effect is transmitted from Tyr to Lys/Arg side-chains. In the discussion they only consider a shielding effect, suggesting that the ligand hampers the reaction through a steric effect (i.e. covering the protein surface), but do not consider a possible structural effect of pinocembrin, which could affect the protein conformation. This possibility could be even controlled by CD spectroscopy, by comparing the spectrum of the protein alone and in the presence of pinocembrin, or at least addressed in the discussion.
3. The shielding of lysine/arginine side-chains upon pinocembrin binding (see previous point) should selectively inhibit the glycation of residues close to Tyr in the sequence/structure. The authors should compare the results of the glycation with and without pinocembrin to understand if the inhibitory effect is selective or not. Are the glycation sites of native insulin known? How many Lys/Arg are glycated out of the total in the native protein? Are there any Arg/Lys residues adjacent to Tyr residues? I suggest to address these aspects in the discussion, trying to depict a molecular mechanism and making more general its meaning and potential applications.
4. The authors should indicate if there are other dietary flavonoids with similar activity, and also discuss the possibility of a common action mechanism. Furthermore, they should mention some other proteins whose structure/function are impaired by glycation, including Ab peptides (Jana AK, et al., Phys Chem Chem Phys. 2016, 18; Emendato A et al., J. Biol. Chem. 2018, 293), whose interaction with RAGE is quoted in the introduction (pag. 2 lines 45-49) and in the discussion (pag. 12, line 322), and even the impairment of glucose metabolism and production of AGE in the brain of AD patients, which has been very recently reviewed by Butterfield and Halliwell (Nat Rev Neurosci. 2019, 3).
Some other minor comments/suggestions:
-Insertion of a figure with the chemical structure of pinocembrin could help.
-In the abstract define RAGE, ROS, Ab etc. … before the respective abbreviation.
-Pag. 3, lines 94-96: add details and or reference about insulin preparation in the monomeric state, i.e. buffer (if any) and controls (gel-filtration? PAGE?)
Author Response
Enclosed please find a file with point by point response.

Reviewer 3 Report
Title: Pinocembrin protects from the age-induced cytotoxicity and inhibits the non-enzymatic glycation in human insulin
General Comments
The submitted manuscript investigates on the role of pinocembrin on AGE-induced cytotoxicity and inhibition of glycation related receptors. The study attempts to contribute towards the scientific literature on the role of natural products on oxidative damage and AGE biomarkers.
A few clarification and suggestions to the authors to address before this article could be accepted for publications.
Title: suggest the authors replace “age-induced” with “AGE-induced’ if referring to advanced glycation endproduct as the latter is more logical
Abstract: line 10-11 is not reading correct. Suggest to replace “as associated” to “for its association”
Introduction: Line 74, as the authors made attempts to link with their previous study [36] this needs to be clearly stipulated. The finding from study [36] is linked with the current study as seen in the results and in discussion thus the sentence on line 74-76 needs to be revised and correctly reflected. This will allow the readers to understand that the current study is a follow up on the pathways studies in the previous study [36].
Results
I have major concerns on how the results are reflected in this study and suggest the authors the following.
1. The authors have indicated a two-tailed t-test and significance set at 5% however, none of the results indicates any statistical data. In addition, the authors are requested to perform a posthoc analysis or multiple comparison as there were more than 2 treatments as per each experiment. Data on Figure 1, 2, 3, 4, and 5 to include multiple comparison data.
2. With the current format of results, it is difficult to comprehend why pinocembrin was used as 1:1 ratio with either 40 µM or 10µM for different studies. The authors need to rationalize as to why these concentration are used. In cell based studies it is expected that a dose depend response is reflected including concentration which are physiological relevant. This is the missing link which the authors needs to address.
3. Reading further on (Section 3.3 and Section 3.4) of the results, some of the concerns raised in point 2 above may be address. Suggest that the authors move the insulin glycation reaction (Section 3.3) and characterization (Section3.4) to section 3.1 and 3.2 respectively. As these sections includes some rationale for the 1:1 ratio (section 3.4) including the concentration depended (section 3.3) studies. However, the authors are requested to provide rationale for the 10 µM or 40 µM concentration used in the cell-based studies.
4. Line 193-197, (and throughout the result section) discusses the results and the authors are making a bold statement of significant changes in the presence of pinocembrin however there is no statistical data presented. The authors are suggested to check this and provide the statistical data if there is a confirmed significant difference observed.
5. Under each sub-section, the authors have just made general observation with reference to the respective figures. As there is limitation in the statistical data its difficult to follow the trends and any changes seen with the incorporation of the inhibitor. The authors may want to compare the observation with respective control before reflecting those in the result section
6. Figure 1; Panel B details is missing. In addition the information on p<0.01 is meaningless as the data presented needs multiple comparison not a t-test only.
7. Figure 2: reflect how the data is presented ie mean ± SD or SEM. This needs to be checked for all the figures. Also the stats are missing which needs to be reflected for other figures as well.
8. Figure 4 and 5, the data presented is hard to follow. It is very likely that the authors have used the same control experiment thus the data of A and B can be combined to better reflect the trends. This could be done using line graph rather than bar-graph. This will also allow the authors to compare the reactivity of D-ribose and methylglyoxal. Tukey’s Multiple comparison test may be more suitable for these sets of data. This will also allow to expand on the results presented in line 266-269
Discussion: the authors made good attempt to discuss the results and brings in adequate literature information to support the findings. Perhaps the authors might want to consider some minor suggestions:
1. Line 332-336: Make study [36] as stronger rationale for undertaking the current study. Possible statements such as “in our previous study or we have previously shown…….”.
2. Line 350, reference needed
3. Line 373, the phrase “socio-economic impact” is confusing, are the authors referring to the disease burden ie physiological and pathological significance of AGEs.
4. Throughout the manuscript the authors have used the linking statement as “indeed”. This is too often used in this manuscript which is not needed and distracting the readers. Authors are suggested to revise this and delete where its not needed in a scientific manuscript. These are too “verbose”
General remark: Result sections needs major revision with statistical data and should better accounted for in the result trends and observation as the current description are too generic. These needs to be clearly reflected before the manuscript could be accepted for publication
Author Response

(The authors gave the same response as above.)

Reviewer 4 Report
This study is innovative and important because the research of anti-glycation compounds is one of the strategies for limiting aging-related
chronic disease.
It is very well-written, and a very well performed experiment with no significant flaws, I just have a comment:
In the statistical analysis section, they mentioned that
"For statistical analysis, we used a two-tailed Student’s t-test with unequal variance at a significance level of 5% unless otherwise indicated"
However, in figure 1 panel A they presented results from Cell viability evaluated by MTT assay in the presence and in the absence of pinocembrin. They should consider the use of an ANOVA test to perform this analysis.
Author Response

(The authors gave the same response as above.)

Round 2
Reviewer 1 Report
In the new version of the manuscript, the authors have included several modifications, which resulted into an improved version of the manuscript. However, I still find several issues that precluded its acceptance.
1-In my previous report, I asked the authors to complete the Figure 1, 2 and 3D,E with the controls of ribose and ribose+pinocembrim. Instead of providing them, they just said that the samples were lyophilized. I have no doubt about it, but I still find the need of providing such controls since this information is crucial to understand whether the observed effect comes from the ribose moiety itself or from a reaction product formed between insulin and ribose.
2- The authors neither provided the images showing the cells treated only with ribose. I do not really understand why they do not provide the information that I requested and they just reply my query providing information that I did not ask for. Can the authors provide the confocal images showing the cells treated with ribose alone?
6-In my previous report I asked to provide experimental evidences showing that the MG that they used was pure. Instead of providing them, they just said that it was. I am quite amazed about their answer. However, they insisted that they have the LC-MS analysis. Hence, in their rebuttal they must show this information (not need to include it in the manuscript).
7.-Again, I asked the authors to provide the fluorescent spectra of pinocembrin (at λexc 275nm and 320nm). Instead of providing it, the authors just said that the fluorescent signal was negligible. I do not understand why the authors do not show this data. In addition, I have strongly doubts about their assessment, since pinocembrin is able to strongly absorbs UV-Vis radiation at 275nm [1].
[1] Alimkhodzhaeva G. Kh. et al., Quantitative analysis of pinocembrin. Chemistry of Natural Compounds, 30, 411-413 (1994).